# A New Topical Candidate in Acne Treatment: Characterization of the Meclozine Hydrochloride as an Anti-Inflammatory Compound from In Vitro to a Preliminary Clinical Study

**DOI:** 10.3390/biomedicines10050931

**Published:** 2022-04-19

**Authors:** Philippe A. Grange, Guillaume Ollagnier, Laurianne Beauvais Remigereau, Carole Nicco, Constance Mayslich, Anne-Geneviève Marcelin, Vincent Calvez, Nicolas Dupin

**Affiliations:** 1Département DRC, Développement, Reproduction et Cancer, Institut Cochin, INSERM U1016-CNRS UMR8104, Université de Paris Cité, 75014 Paris, France; philippe.grange@aphp.fr (P.A.G.); guillaume.ollagnier@inserm.fr (G.O.); constance.mayslich@inserm.fr (C.M.); 2Service de Dermatologie-Vénéréologie et CeGIDD, Groupe Hospitalier APHP.5, CNR IST Bactériennes—Laboratoire Associé Syphilis, 75014 Paris, France; 3SkinDermic, 75014 Paris, France; lauriannebr@lbrconseil.fr (L.B.R.); anne-genevieve.marcelin@aphp.fr (A.-G.M.); vincent.calvez@aphp.fr (V.C.); 4Département 3I, Infection, Immunité et Inflammation, Institut Cochin, INSERM U1016-CNRS UMR8104, Université de Paris Cité, 75014 Paris, France; carole.nicco@inserm.fr; 5Service de Virologie, Hôpital Pitié Salpêtrière, APHP, Institut Pierre Louis d’Epidémiologie et de Santé Publique (iPLESP), INSERM UMR_1136, Sorbonne Université, Virologie, 75013 Paris, France

**Keywords:** acne, *C. acnes*, inflammation, meclozine, anti-histaminic, human testing

## Abstract

Acne is a chronic inflammatory multifactorial disease involving the anaerobic bacterium *Cutibacterium acnes* (*C. acnes*). Current acne treatments are associated with adverse effects, limiting treatment compliance and use. We showed that meclozine, an anti-histaminic H1 compound, has anti-inflammatory properties. In Vitro, meclozine reduced the production of CXCL8/IL-8 and IL-1β mRNA and protein by *C. acnes*-stimulated human keratinocytes and monocytes. No cell toxicity was observed at the IC50. Meclozine prevented the phosphorylation of ERK and JNK. In Vivo, 1% meclozine gel significantly decreased *C. acnes*-mouse ear induced inflammation by 26.7% (*p* = 0.021). Ex vivo experiments on human skin explants showed that meclozine decreased the production of GM-CSF, IL-1β and TNF-α at transcriptional and translational levels. In a randomized, double-blind, placebo-controlled proof-of-concept clinical trial on 60 volunteers, 2% meclozine pharmaceutical gel decreased by 20.1% (*p* < 0.001) the ASI score in the treated group after 12 weeks of treatment. No adverse event was reported. Together, these results indicate that meclozine is a potent topical anti-inflammatory compound of potential value for acne treatment.

## 1. Introduction

Acne vulgaris is a disorder affecting the pilosebaceous unit (PSU), resulting in both inflammatory and non-inflammatory clinical lesions. Acne may range from mild symptoms to a severe rare fulminant presentation in which patients display a highly destructive inflammatory response that is often associated with scarring [1]. However, most patients display a combination of both non-inflammatory comedones and inflammatory papules, pustules and nodules. It is estimated that 80–85% of teenagers are affected by acne, which may persist into adulthood [2]. Acne is not life-threatening but may have serious physical and psychological consequences if left untreated.

Acne is a multifactorial inflammatory disease involving excess sebum production, which induces the proliferation of keratinocytes, leading to obstruction of the follicle, establishing hypoxic conditions favoring the growth of *Cutibacterium acnes* (*C. acnes*, previously known as *Propionibacterium acnes*) and inducing inflammation [3,4]. *C. acnes* phylotype IA_1_ is strongly associated with inflammatory acne. Recent studies have shown that *C. acnes* plays a key role in acne, but that imbalances in the cutaneous microbiota favor the selection of pathogenic *C. acnes* strains of a specific lineage capable of producing several virulence factors (biofilm, surface proteins), thereby increasing inflammatory capacity [5,6,7,8,9,10,11]. *C. acnes* appears to be an opportunistic pathogen, as it has also been implicated in other inflammatory diseases, implant-associated infections, sarcoidosis, synovitis-acne-pustulosis-hyperostosis-osteitis (SAPHO) syndrome and prostate cancer, in which phylotypes IB, IC, II and III are most frequently identified [12,13,14,15,16].

Moreover, *C. acnes* can induce inflammatory response by activating the innate immune system via the Toll-like receptors TLR2 and TLR4, and by activating the NF-κB, MAPK and NLRP3 inflammasome signaling pathways, leading to production of the proinflammatory molecules IL-1α/β, IL-6, CXCL8/IL-8, IL-12, granulocyte-macrophage colony-stimulating factor (GM-CSF), TNF-α, β-defensin-2 (hBD-2), and matrix metalloproteases (MMPs) by keratinocytes, sebocytes and monocytes in vitro and ex vivo, in acne lesions [17,18,19,20,21,22,23]. *C. acnes* CAMP factor 1 is also involved in the production of CXCL8/IL-8 via TLR2 in keratinocytes and of type I interferon (IFN-I) via the cGAS-STING pathway in macrophages [24,25].

The treatments targeting the underlying causes of acne developed, to date, have focused on sebum production, keratinocyte proliferation, the bacterial population and inflammation in the PSU. Depending on the severity of the acne, these treatments are administered topically or systematically, as single agents or in combinations. The agents used include topical retinoids, topical and/or oral antibiotics directed against *C. acnes*, including erythromycin, clindamycin, or doxycycline and diaminodiphenyl sulfone (DDS or dapsone), oral contraceptives and isotretinoin [26]. Anti-inflammatory agents, such as salicylic acid and benzoyl peroxide (BPO), can reduce inflammatory lesions in acne, but are associated with adverse effects, such as redness, burning, irritation and allergic contact dermatitis. Retinoid agents, such as adapalene, tretinoin, isotretinoin, and tazarotene, are comedolytic; they normalize desquamation in the PSU, and have anti-inflammatory properties. Acne responds very slowly to antibiotic treatments, the widespread use of which has led to the development of resistance to macrolide antibiotics and tetracyclines in *C. acnes*, with 40% of strains now resistant to these agents [27,28]. Several lines of research are now focusing on the development of new therapeutic approaches.

In this study, we identified meclozine dihydrochloride—which has antihistamine activity, widely used as an anti-emetic drug—as a new anti-inflammatory molecule. Meclozine efficiently decreased *C. acnes*-induced inflammation in primary keratinocytes and monocytes in vitro, in vivo in the mouse ear inflammation model, and ex vivo in human skin explants. It also significantly decreased inflammatory acne lesion count in a proof-of-concept clinical trial.

## 2. Materials and Methods

### 2.1. Chemicals

Meclozine, CAS number 31884-77-2, MTT reagent, and DMSO were purchased from Sigma (Merck). Larger quantities of meclozine were purchased from LGM (LGM Pharma, LLC, Erlanger, KY, USA).

### 2.2. Cell Culture, Pre-Treatment and Stimulation

Immortalized human keratinocytes cells (HaCaT, GmbH, CLS Cell Lines Services GmbH, Eppelheim, Germany; catalog number 300493) were grown in DMEM- Glutamax-I supplemented with 1 mM sodium pyruvate. The immortalized human monocytic cells ThP-1 (ATCC, Manassa, VA, USA; catalog number TIB-202) were grown in RPMI-Glutamax-I. DMEM and RPMI were supplemented with 0.1% and 10% heat-inactivated fetal calf serum (ThermoFisher Scientific, Life Technologies SAS, Courtaboeuf, France), and with an antibiotic/antimycotic solution (10 U/mL penicillin, 10 μg/mL streptomycin, 0.25 μg/mL amphotericin (Life Technologies)). Primary human epidermal keratinocytes (NHEK) were grown in the KGM-Gold Bullet Kit, in accordance with the manufacturer’s instructions (Lonza, Basel, Switzerland). Immortalized cell lines were routinely checked for the absence of *Mycoplasma* infection. All cells were incubated at 37 °C in a humidified atmosphere containing 5% CO_2_. Cultured cells in 6- or 96-well polystyrene plates were treated with meclozine solution for 1 to 48 h at 37 °C. After changing culture media, cells were stimulated with a *C. acnes* suspension adjusted to the appropriate concentration (see below) for 30 min to 18 h at 37 °C, under an atmosphere containing 5% CO_2_.

### 2.3. Bacterial Strain and Growth Conditions

*C. acnes* strain 6919 (Type IA_1_) was obtained from the American Type Culture Collection (Manassas, VA). *C. acnes* strains RON (IA_1_) and PIE (IB), isolated from a patient with joint infection, were kindly provided by Dr. Philippe Morand from the Bacteriology Department at Cochin Hospital, Paris, France [24]. All strains were cultured under anaerobic conditions, in liquid or solidified reinforced clostridial medium (RCM) (Difco Laboratories, Detroit, MI), with a GasPak™ EZ Anaerobic Container System (Becton Dickinson & Co, Sparks, MD, USA). We used 100 mL RCM for routine culture, with bacteria harvested after five days at 37 °C, by centrifugation at 7000× *g* for 10 min at 4 °C. Pellets were pooled and washed in about 30 mL cold sterile PBS [1.5 mM KH_2_PO_4_, 2.7 mM Na_2_HPO_4_.7H_2_O, 0.15 M NaCl (pH 7.4)] and were then centrifuged again, as described above. Finally, the bacterial pellet was suspended in sterile PBS (1:10 from volume culture).

### 2.4. Screening of 2145 FDA-Approved Compounds

The molecules tested were selected from chemical and plant libraries containing 1794 and 351 molecules (Prestwick 14D1305 MP02 ST05 and 02-Nat. Mp01 D1203 ST08), respectively. They were distributed between the wells of twenty 96-well plates at a concentration of 10 mM, and stored at −80 °C. Before use, the plates were thawed in the dark to distribute the initial volume in 4 × 25 µL. Primary human keratinocytes cells (NHEK) were used to seed a 96-well culture plate at a density of about ~5 × 10^4^ cells per well and were incubated for 24 h in the presence of chemical compounds at a concentration of 10 µM. After changing culture media, they were stimulated with a *C. acnes* suspension (MOI of 15) for 18 h. CXCL8/IL-8 levels in the culture supernatant were determined by ELISA and cell viability was estimated with the MTT assay as described below.

### 2.5. Cell Viability Assays

Cell viability was estimated in MTT assays, in which cells were incubated with a 0.2% MTT solution in cell culture medium for 4 h at 37 °C. This solution was then removed and DMSO was added to solubilize the MTT-formazan compound produced in living cells. After thorough mixing, absorbance was measured at 550 nm.

### 2.6. ELISA

The concentrations of human IL-1β and CXCL8/IL-8 proteins in the supernatants of stimulated cells were determined with Ready-Set-Go ELISA Sets (Thermo Fisher Scientific, Life Technologies SAS, Courtaboeuf, France) according to the manufacturer’s instructions. We used serial dilutions of recombinant human IL-1β and CXCL8/IL-8 to generate a standard curve. We determined optical density at 450 nm, with wavelength correction at 570 nm.

### 2.7. Luminex Assay

ProcartaPlex human 3-plex assays (Thermo Fisher Scientific, Life Technologies SAS, Courtaboeuf, France) were used to determine cytokine levels (GM-CSF, IL-1β/IL-1F2, TNF-α) in the supernatant of *C. acnes*-infected and uninfected cells and skin explants, according to the manufacturer’s instructions.

### 2.8. Quantitative Real Time Polymerase Chain Reaction (qRT-PCR)

Cells and skin explants were cultured in 6- to 12-well polystyrene plates and were subjected to a 24–48 h pre-treatment with 10 µM meclozine before stimulation for 5–18 h with *C. acnes*, as described above. Total RNA was isolated with NucleoSpin RNA kits and treated with DNAse I, according to the manufacturer’s instructions (Macherey-Nagel, Hoerdt, France). RNA concentration was determined at 260 nm on a Nanodrop spectrometer (Labtech, France) and the A_260_/A_280_ ratios for all samples ranged between 1.6 and 1.9. Complementary DNA was generated from 100 ng of total RNA, by reverse transcription at 50 °C for 10 min. The cDNA was then subjected to quantitative PCR analysis, in the QuantStudio™ 5 Real-Time PCR System thermocycler (Applied Biosystems) with the iTaq Universal SYBR Green One-Step kit (Bio-Rad Laboratories, Hercules, CA, USA). A two-step cycling system was used: 95 °C for 60 s followed by 40 cycles of 95 °C for 15 s, 68 °C for 60 s, and a final melting curve determination from 65–95 °C during 60 s, with 0.1 °C/s increments. The threshold cycle (Ct) was determined from the amplification curves for the genes studied. The amount of RNA in stimulated cells relative to control cells was calculated by the 2ΔCt method and is expressed as a relative fold-change in expression, normalized against the expression level of an internal control gene (GAPDH). All the primers used in this study are summarized in Appendix A.

### 2.9. Immunoblotting Analysis

HaCaT cells were cultured in 6-well polystyrene plates and were subjected to a 24 h pre-treatment with 10 µM meclozine before stimulation for 30 min to 3 h with *C. acnes*, as described above. HaCaT cells were washed in cold sterile PBS and scraped into RIPA lysis buffer containing 50 mM Tris (pH 8.0), 150 mM NaCl, 1% NP-40, 0.5% sodium deoxycholate, 0.1% SDS, 2 mM EDTA, 2 mM sodium pyrophosphate,10% glycerol, 1 mM sodium orthovanadate, 1 mM phenylmethylsulfonyl fluoride and 10 mg/mL leupeptin. The lysate was centrifuged at 14,000× *g* for 15 min at 4 °C and the protein concentration of the resulting total protein extract was determined by the Lowry method with BSA as the standard, as described by Peterson [29]. Proteins (30 µg per lane) were fractionated by electrophoresis (LDS-PAGE) under denaturing conditions in a NuPAGE Novex 4–12% Bis-Tris gel (1 mm, 12 wells, Invitrogen, UK) and the resulting bands were transferred onto nitrocellulose membranes. The membranes were saturated by incubation for 1 h at room temperature in 5% nonfat milk powder in TBS/T buffer [1 × TBS, 0.1% Tween-20] and incubated for 18 h at 4 °C with rabbit polyclonal IgG antibodies against p-p38 (Catalog #sc-17852-R, Thr 180/Tyr 182, 1:1000 BSA), p38 (Catalog #sc-535, C-20), ERK1 (Catalog #sc94, K-23), JNK1/2 (Catalog #sc-571, FL, 1:1000 BSA), p-PKC (Cell Signaling, Catalog #9371, pan bII Ser660), IkB (Catalog #sc-371, C-21), and mouse monoclonal IgG antibodies against native and phosphorylated forms of p-ERK (Catalog #sc-7383, E-4), p-JNK (Catalog #sc-6254, G-7), PKC (Catalog #sc-17769, A-3), β-actin (Catalog #sc-47778, C4) used as a protein loading control, and goat polyclonal IgG antibody against p-IkBa (Catalog #sc-7977, Ser 32) diluted to 1:1000 in 5% BSA in TBS/T buffer (all antibodies were purchased from Santa Cruz Biotechnology, Inc., Santa Cruz, CA, USA, unless otherwise stated). The membranes were washed and the bound antibodies were detected by incubation for 1 h at room temperature with polyclonal goat anti-mouse IgG-HRP (Catalog #sc-2005), goat anti-rabbit IgG-HRP (Catalog #sc-2004), donkey anti-goat IgG-HRP (Catalog #sc-2020) antibodies diluted 1:2000 in saturation buffer. Unbound material was removed by washing and peroxidase activity was detected by chemiluminescence with ECL Western Blotting Reagent (Advansta Corp., Menio Park, CA, USA).

### 2.10. C. acnes-Induced Inflammation in the Mouse Ear Model In Vivo

Each group of mice studied contained eight 6-week-old female BALC/c mice purchased from Janvier Laboratory (Le Genest Saint Isle, France). The mice in the negative control group (PBS) were untreated and received intradermal injections of PBS (20 µL) into both ears; the mice in the positive control group (*C. acnes* + vehicle) received intradermal injections of *C. acnes* suspension (20 µL of a 10^6^ bacteria/mL suspension) and topical treatment with vehicle-gel alone (30 mg each ear); the mice in the treated group (meclozine-treated) received intradermal injections of *C. acnes* suspension into each ear and were then treated topically with meclozine gel (30 mg each ear). For the in-house gel preparation, meclozine was extemporaneously dissolved in 150 µL DMSO, and the resulting solution was mixed with HS153070/H_2_O (30:70, *w*/*w*) and incorporated into Vaseline to achieve the desired concentration (0.1, 0.5, 1, 2 or 4%). For the pharmaceutical gel preparation, the formulation contained 50% propylene glycol, 47% purified water, 1.5% hydroxyethylcellulose (Natrosol 250 XHP), 0.5% benzyl alcohol and 1% meclozine (Eurofins Amatsigroup, Saint Gély du Fesc, France). For both gel preparations, the vehicle consisted of all the ingredients except meclozine. The local inflammatory reaction was scored daily over a period of four days after *C. acnes* injection. For each ear, we evaluated the following parameters: thickness, with a micro caliper (Mitutoyo, Kawasaki, Japan), redness and the presence of desquamation and/or pustules. Ear thickness was measured in millimeters and converted into a score in arbitrary units (a.u.) as follows: 0–0.20 mm = 0 a.u.; 0.21–0.40 mm = 0.5 a.u.; 0.41–0.60 mm = 1.0 a.u.; 0.61–0.80 mm = 1.5 a.u.; 0.81–1.0 mm = 2.0 a.u.; 1.01–1.2 mm = 2.5 a.u.; 1.21–1.40 mm = 3 a.u. Redness, desquamation and/or the presence of small pustules were scored as follows: low = 0.5 a.u., moderate = 1.0 a.u., high = 1.5 a.u. and very high = 2.0 a.u. The inflammation score for a single mouse corresponded to the mean of all arbitrary-unit scores obtained for each ear. At the end of the experiment, a final inflammatory score was obtained and the ears were photographed. The mice were killed and their ears were removed and fixed in a formalin-containing buffer for further histological analysis.

### 2.11. C. acnes-Induced Inflammation Assay Ex Vivo

Full-thickness human skin was obtained from excess tissue removed during abdominoplasty procedures at the De La Tour Clinique in Paris, France. Tissue collection was anonymous, in accordance with routine procedures approved by the local ethics committee. Written informed consent was obtained from all patients. Subcutaneous fat was trimmed from the skin tissue. Punch biopsies (8 mm) were performed and the explants obtained were placed immediately in culture medium containing DMEM/RPMI/SVF (50/50/20) in the presence of 2% penicillin/streptomycin and 1% fungicide, supplemented with 250 µg/mL insulin, 0.1 mg/mL EGF and 0.1 mg/mL FGF. They were incubated at 37 °C under an atmosphere containing 5% CO_2_ for 18 h to check for the absence of bacterial contamination. Skin explants were then subjected to pre-treatment with 15 µM meclozine for 48 h at 37 °C, followed by stimulation with a *C. acnes* suspension (O.D._600nm_ = 1.0) for 18 h at 37 °C. A baseline control was realized with an untreated and unstimulated skin explant. The positive control corresponded to a skin explant without pre-treatment but stimulated with the *C. acnes* suspension.

### 2.12. Clinical Trial

#### 2.12.1. Study Design

The study was a 12-week, randomized, double-bind, placebo-controlled study under dermatological control.

#### 2.12.2. Efficacy Assessment

Michaelson’s acne severity index (ASI), which reflects acne severity, was the key assessment in this study. It was evaluated by counting non-inflammatory lesions (open and closed comedones) and inflammatory lesions (papules, pustules and nodules of at least 5 mm in diameter), and calculated as follows: ASI = (comedones × 0.5) + (papules × 1) + (pustules × 2) + (nodules × 3).

#### 2.12.3. Safety Assessments

Throughout the study, safety was assessed through physical examinations by a dermatologist or by the recording of patient-reported adverse events at each visit.

#### 2.12.4. Study Product

Both, the meclozine and vehicle gels were manufactured by Marie Pratt Cosmetic (Marie Pratt Cosmetic, Viry Châtillon, France) and directly shipped to South Korea for the study. The test product consisted of 2% meclozine gel in a 30 mL plastic tube. It was tested to check for the absence of bacterial contaminants before the start of the study. The vehicle was of identical composition, except that it contained no meclozine.

#### 2.12.5. Participants

All the acne patients studied were included by a dermatologist. After one week of inclusion (D0), the volunteers were randomly selected into two groups of 30 volunteers each, to receive either the active formulation containing 2% meclozine or the vehicle control. For allocation, a random computer-generated list was used. The study was conducted as a double-blind trial in which neither researchers nor the volunteers knew which was test product or placebo. These products were applied to the face twice, daily, morning and evening (at bedtime) on a clean and perfectly dry face after the application of a toner. The exclusion criteria were as follows: (1) severe chronic wasting disease; (2) allergies, sensitive or hypersensitive skin; (3) skin disorders or skin diseases other than acne at the test site; (4) the use of functional cosmetics for acne during the two weeks before the start of the study; (5) known or suspected contact sensitization to the ingredients of the test formulation; (6) any treatment for acne during the four weeks before the start of the study; (7) pregnancy, breastfeeding or plans to conceive; (8) use of oral contraceptives during the three months before the start of the study; (9) use of oral retinoids during the six months before the start of the study or of any treatment likely to affect sebum secretion in the long term; (10) a prominent nutritional disorder; (11) drug or alcohol dependence; (12) presence of a tattoo, scars, erythema, telangiectasis, or burns at the test site; (13) participation in other clinical studies or other clinical testing bodies.

#### 2.12.6. Clinical Test Procedure and Outline

Acne patients were asked to attend seven visits, at inclusion (baseline), intermediate time points (at 2, 4, 6, 8, and 10 weeks) and at the end of the study (12 weeks). At baseline, inflammatory and non-inflammatory lesions were counted to determine ASI (see Section 2.12.2). The facial images were photographed by VISIA-CR in a cross-polarized mode (Canfield Imaging System, Canfield Scientific, Parsippany, NJ, USA). During intermediate visits and the end-of-study visit, inflammatory and non-inflammatory lesions were counted and facial images were again obtained. Compliance was checked by retrieving the remaining test produce and determining the amount used, and adverse skin reactions were assessed.

### 2.13. Statistical Analysis

The statistical significance of differences between experimental groups was determined by using parametric unpaired *t*-tests and ANOVA test. For in vivo study unpaired, a non-parametric Mann–Whitney test was used. For the clinical study, data in intention-to-treat (ITT) and per-protocol (PP) were analyzed and the statistical significance of differences between groups was assessed in Wilcoxon tests to evaluate the effect of meclozine treatment over time, between baseline and 12 weeks. Mann–Whitney tests were used to determine the significance of differences between the two products from baseline to the visit at 12 weeks. All calculations were made with GraphPad Prism 9 (GraphPad, La Jolla, CA, USA). A *p*-value ≤ 0.05 was considered statistically significant, and the level of significance is indicated as follows: * (*p* ≤ 0.05), ** (*p* ≤ 0.01), and *** (*p* ≤ 0.001).

## 3. Results

### 3.1. Chemical Library Screening

In screening assays, all molecules were tested at a concentration of 10 µM on human primary keratinocytes (NHEK), which were pretreated for 24 h before *C. acnes* stimulation for 18 h. We found that 1320 of the 2145 molecules tested (61%) had a toxicity of no more than 20%. Eight of these 1320 molecules (0.6%) decreased CXCL8/IL-8 production by at least 70%. One of these eight preselected molecules, the meclozine hydrochloride (or meclozine), decreased CXCL8/IL-8 production by 86% and was selected for further study (Figure 1).

### 3.2. Meclozine Inhibits the C. acnes-Induced Production of CXCL8/IL-8 and IL-1β in Keratinocytes and Monocytes

The anti-CXCL8/IL-8 activity of meclozine was confirmed by pre-treatment for 24 h of the keratinocyte cells (HaCaT and human primary NHEK) and human monocyte cells (ThP-1) with concentrations ranging between 0.39 to 50 µM, followed by the stimulation with *C. acnes* suspension. Meclozine inhibited the production of CXCL8/IL-8 in both human primary and immortalized keratinocyte cells in a dose-dependent manner, with an IC_50_ of about 6–7 µM (*p* < 0.0001 for HaCaT and *p* = 0.0132 for NHEK) (Figure 2A,D). No cytotoxicity of the meclozine was observed at the IC_50_ in either of the cell lines (Figure 2B,E). Moreover, meclozine pre-treatment decreased CXCL8/IL-8 mRNA levels by 88.7%, relative to the untreated control in HaCaT cell (Figure 2C). In the human monocyte cells (ThP-1), meclozine inhibited the production of IL-1β protein in a dose-dependent manner, with an IC_50_ of about 12 µM (*p* = 0.0035) (Figure 2F). No cytotoxicity was measured (Figure 2G). Moreover, IL-1β mRNA production decreased by 45% in treated cells (Figure 2H). Similar results were obtained with two other *C. acnes* strains (RON phylotype IA_1_ and PIE phylotype IB) for keratinocytes (HaCaT and NHEK) and monocytes (ThP-1) (Appendix A). Finally, we evaluated the effect of various durations of meclozine pre-treatment on CXCL8/IL-8 production. HaCaT cells were subjected to meclozine pre-treatment at concentrations of 0.39 to 50 µM for 1, 6, 24 and 48 h and were then stimulated with *C. acnes*. Pre-treatment with meclozine for 1 or 6 h did not significantly decrease CXCL8/IL-8 production (Figure 2I). However, after 24 h of pre-treatment, meclozine had a clear effect, decreasing CXCL8/IL-8 production by 52.7% when used at a concentration of 6.25 µM (*p* = 0.0018). This effect was further enhanced by increasing the duration of pre-treatment to 48 h, which resulted in a 75% decrease in CXCL8/IL-8 production (*p* < 0.001) (Figure 2I). At the IC_50_, no significant impact on cell viability was measured at any pre-treatment time duration. However, at the IC_90_, the viability of cells decreased significantly after 48 h pre-treatment at high meclozine concentrations (25 and 50 µM) (Figure 2J).

### 3.3. Meclozine Inhibits the PGN- and LTA-Induced Production of CXCL8/IL-8 and IL-1β

Gram-positive bacteria have a thick layer of peptidoglycan (PGN), in which lipoteichoic acid (LTA) is embedded in the cell envelope [30]. We therefore investigated whether meclozine decreased the production of pro-inflammatory molecules following cell stimulation by using both components. HaCaT and ThP-1 cells were subjected to pre-treatment with concentrations of meclozine ranging from 0.39 to 25 µM for 24 h. They were then stimulated with PGN and LTA at concentrations of 5, 10 and 20 µg/mL, for 18 h at 37 °C. In HaCaT cells, CXCL8/IL-8 production increased in a dose-dependent manner in cells stimulated with PGN alone, from 70 pg/mL to up to 130 and 180 pg/mL. Pre-treatment with 6.25 µM meclozine decreased the production of CXCL8/IL-8 by a mean of 60%, regardless of the initial PGN dose used (Figure 3A). Cells stimulated with LTA produced very small amounts of CXCL8/IL-8, but these amounts were decreased by 55–65% following pre-treatment with 6.25 µM meclozine (Figure 3B). In ThP-1 cells, PGN alone induced a slight dose-dependent increase in IL-1β production, which was 30 to 45% smaller following pre-treatment with 6.25 µM meclozine (Figure 3C). However, small amounts of IL-1β were produced after LTA stimulation, and these amounts were not significantly affected by pre-treatment with 6.25 µM meclozine (Figure 3D).

### 3.4. Molecular Mechanism of Action of Meclozine

We investigated the signaling pathways affected by the pre-treatment of HaCaT keratinocytes with meclozine before their stimulation with *C. acnes*. We first confirmed the time-dependent production of CXCL8/IL-8 in the presence of *C. acnes*, and its abolition by meclozine pre-treatment (Figure 4A). The stimulation of HaCaT keratinocytes with *C. acnes* was associated with the degradation of p-IκB while total IκB did not change (Figure 4B,D). In presence of meclozine, while the overall signal is low, the degradation of p-IκB was still present (Figure 4C,D). When analyzing the MAPK pathway, we showed that *C. acnes* induced a transient activation of pERK and pJNK, and in lesser extent, p-p38 (Figure 4B,E–G). Pre-treatment of HaCaT keratinocytes with meclozine abolished the phosphorylation of ERK and JNK (Figure 4C,E,G), while the phosphorylation of p38 was not impacted (Figure 4C,F). The stripping and subsequent re-probing of the blot with antibodies against total ERKs, p38s, and JNKs revealed an absence of change in total protein levels following *C. acnes* stimulation. It is known that meclozine interacts with the H1-receptor [31]. We therefore evaluated its effect on the PKC. In the presence of *C. acnes*, we observed a phosphorylation of PKC after 2 h of *C. acnes* stimulation. In presence of meclozine, the phosphorylation of PKC still remained but appeared to be slightly delayed, taking place after 3 h of *C. acnes* stimulation (Figure 4B,C,H). These data suggest that the inhibition of *C. acnes*-induced CXCL8/IL-8 production in keratinocytes by meclozine involves mainly the downregulation of the MAPK pathway through ERK and JNK.

### 3.5. Meclozine Reduces C. acnes-Induced Inflammation In Vivo

Meclozine decreases the *C. acnes*-induced production of cytokines and chemokines in vitro. We therefore evaluated its anti-inflammatory capability in vivo, in a mouse ear inflammation model. We first assessed the effect of meclozine incorporated into an in-house Vaseline-based cream at concentrations ranging from 0.1 to 4% (Figure 5A). The injection of the *C. acnes* suspension into the ears of mice treated with the vehicle resulted in significant inflammation, characterized by an increase in inflammatory score to levels 3.8 times higher than for the PBS negative control group within 96 h (Figure 5B). Treatment of the ears with meclozine gel decreased ear inflammation in a dose-dependent manner (Figure 5B,C). Inflammatory score was 14.3% lower following treatment with 1% meclozine (*p* = 0.020), 50% lower for 2% meclozine (*p* = 0.0011) and 57.1% lower with 4% meclozine (*p* = 0.0005). Histological analysis revealed a large increase in ear thickness associated with a massive inflammatory cell infiltrate relative to the PBS control, in which ear thickness decreased in the presence of meclozine (Figure 5D). We then tested a prototype gel-based pharmaceutical formulation containing 1% meclozine (Figure 5E). In the control group, in which ears were stimulated with *C. acnes* and then treated with the vehicle control, we observed an increase in ear score, as previously described. Ears treated with 1% meclozine gel had a 26.7% lower inflammatory score (*p* = 0.021) than those treated with the vehicle control (Figure 5D), and had no *C. acnes*-related lesions (Figure 5F,G). No inflammation was observed when meclozine was used alone. Overall, these results demonstrate the anti-inflammatory capacity of meclozine in vivo and validate the gel-based formulation for further use in clinical assays. The 1% meclozine gel concentration was considered to be the minimal active concentration.

### 3.6. Meclozine Reduced C. acnes-Induced Inflammation Ex Vivo

We then assessed the ability of meclozine to decrease *C. acnes*-induced inflammation ex vivo, by subjecting human skin explants to pre-treatment with meclozine. Skin explants stimulated with *C. acnes* produced larger amounts of GM-CSF (1373 pg/mL, *p* < 0.0001), TNF-α (46.2 pg/mL, *p* < 0.0001) and IL-1β (79.5 pg/mL, *p* < 0.0001) than unstimulated explants used as baseline controls. In the presence of meclozine, GM-CSF, TNF-α and IL-1β levels were 50.6% (*p* < 0.0001), 74.2% (*p* < 0.0001), and 61.4% (*p* < 0.0001) lower, respectively (Figure 6A,C,E). A similar effect was observed at transcriptional level, with decreases in mRNA levels of 71.7% for IL-1β (*p* < 0.0001), 57.4% for TNF-α (*p* < 0.0001), and 44.6% for GM-CSF (*p* < 0.0001) (Figure 6B,D,F).

### 3.7. Preclinical Toxicological Evaluation

Toxicological assays were performed in accordance with the requirements of cosmetological standards (Appendix A). Meclozine was found to be an ocular irritant, but it was not phototoxic, mutagenic or pro-mutagenic, and it did not sensitize or irritate the skin. We tested pure meclozine on 10 volunteers, to evaluate its cutaneous compatibility. No skin reaction was observed after semi-occlusive application for 48 h (Appendix A).

### 3.8. Clinical Study Proof-of-Concept

We enrolled 60 volunteers (20 men and 40 women) with a mean age of 26.8 years (SD ± 5.86) in the study. Most participants were between 20 and 30 years old. The volunteers were randomized and split into two groups of 30 volunteers (treated and control groups), with a mean age of 27 (SD ± 5.34) and 27 (SD ± 6.43) years, respectively (Table 1). A Mann–Whitney *U* test = 235.5 (*p* = 0.133) showed the two groups to be homogeneous. The treated and control groups corresponded to volunteers receiving the 2% meclozine and the vehicle gels twice daily to the entire face, respectively. Compliance was good, as 78% of the volunteers (47/60) completed the study. Nine of the volunteers in the control group and four of those in the treated group withdrew from the study during the 12-week study period and the reasons for withdrawal were a personal decision to withdraw (6), esthetic reasons (4), overlapping tests (1) and not attending visits (2) (Appendix A). Overall study assessment was made by counting all lesions at baseline and at the end of the 12-week period, and the Michaelson’s acne severity index score was calculated as described in Materials and Methods. In ITT analysis, ASI score decreased by 20.1% (*p* < 0.001) in the treated group (Figure 7A) and by 8.9% (*p* = 0.058) in the control group after 12 weeks of treatment (Figure 7B), corresponding to a difference in percent reduction of 11.2% between both groups. Interestingly, 73.3% of those in the treated group presented a decrease in ASI score, versus only 43.3% of those in the control group (Appendix A). In PP analysis, after meclozine treatment ASI score decreased by 23.3% (*p* = 0.0004) and by 11.3% in the control group, corresponding to a difference in percent reduction of 12% between both groups (Figure 7C,D),. Representative pictures of patients at baseline and after 12 weeks of treatment are shown in Appendix A. No adverse events were reported during the study.

## 4. Discussion

Acne vulgaris is a common chronic disease that is managed with topical and systemic treatments, depending on its severity. Treatments include antibiotics, hormonal, retinoid and anti-inflammatory agents, used alone or in combination. All currently available topical treatments have adverse effects, mostly skin irritation due to the dryness of the skin and allergic contact dermatitis, whereas systemic treatments may cause more serious adverse effects, ranging from skin/mucosal dryness and gastrointestinal problems to liver-function disorders, pancreatitis and psychological distress. Isotretinoin, a very effective treatment for severe acne, is also a potent teratogenic drug with multiple adverse effects subject to restrictions on prescription and prescribed only in the context of pregnancy prevention [27]. There is, therefore, an unmet medical need for new active molecules devoid of adverse effects. Drug repositioning, in which a drug currently used to treat a specific disease is indicated for another disease, has become increasingly important in recent years, particularly in phenotypic drug screening [32]. This strategy has several advantages, including knowledge of the toxicity profile of the drug, minimizing the risk of failure in clinical practice. We performed phenotypic screening with 2145 FDA-approved small molecular compounds, with the aim of identifying a clinically applicable drug capable of improving moderate inflammatory acne in patients. Meclozine was identified as the best candidate. Meclozine dihydrochloride, more generally referred to simply as meclozine or meclizine, is a 1-(*p*-chloro-α-phenylbenzyl)-4-(*m*-methylbenzyl) piperazine dihydrochloride monohydrate compound mostly used as an antihistamine in the management of motion sickness and to treat chemotherapy-induced nausea and vomiting [33]. This anti-inflammatory molecule is well tolerated, has been available over-the-counter for more than 50 years and appears to have few adverse effects. It is largely prescribed in women during pregnancy [34]. We describe here the first study of the effect of meclozine on *C. acnes*-induced inflammation in vitro, in vivo and ex vivo and in a proof-of-concept clinical trial on acne patients.

We found that meclozine decreased the in vitro production of CXCL8/IL-8 and IL-1β mRNA and protein in a dose-dependent manner; CXCL8/IL-8 and IL-1β are known to be involved in acne development [35]. Treatment time appears to be important, as meclozine-induced CXCL8/IL-8 inhibition was effective only after a minimum of 24 h of treatment. Previous studies have shown that the induction, by *C. acnes*, of CXCL8/IL-8 production by keratinocytes involves the NF-κB and MAPK pathways [19,24,36]. We therefore investigated the impact of meclozine on these pathways. We found that meclozine did not inhibit the degradation of p-IκB in the cytoplasm, suggesting that the NF-κB pathway was not impacted by meclozine. The principal MAPKs are p38, ERK and JNK. These proteins are stimulated by TLR2-MyD88, and AP-1 activation, initiated by MAPKs, linked to activation by *C. acnes* [19]. It appears that meclozine did not interfere in the phosphorylation of p38, while the phosphorylation of ERK and JNK were strongly impacted. Overall, meclozine appears to decrease the production of CXCL8/IL-8 by inhibiting the phosphorylation of the ERK and JNK MAPK pathways. These results are consistent with previous findings of an effect of meclozine in the inhibition of inflammation-related osteoclastogenesis by attenuation of the MAPK pathways [37,38] and in the decrease of UV B-induced inflammation in mice [39]. We also found that meclozine decreased the production of IL-1β mRNA and protein in a dose-dependent manner. This result is consistent with reports showing the decrease of IL-1β mRNA production in the kidney in an ischemia-reperfusion injury mouse model after meclozine treatment [40]. Moreover, we showed that meclozine decreased the production of CXCL8/IL-8 when the bacteria cell components PGN and LTA were used in place of *C. acnes* whole bacteria to stimulate keratinocyte cells. This finding is consistent with previous reports showing the production of pro-inflammatory molecules by HaCaT cells stimulated with PGN and LTA [41]. Therefore, meclozine appears to have an anti-inflammatory property not restricted to *C. acnes* but to other cell envelope component-related bacteria, such as PGN and LTA. Further investigations will be necessary to assess the impact of meclozine in the inflammation induced by cutaneous-related gram-positive bacteria.

Meclozine is an H1 histamine receptor (H1R) antagonist that is expressed on keratinocytes and involved in mediating inflammatory responses through cytokine production. H1Rs are G-protein coupled receptors activating the phosphorylation of protein kinase C (PKC), leading to activation of the ERK and NF-κB cascades [42,43]. In this preliminary study, PKC phosphorylation appears to be delayed in the presence of meclozine, suggesting a potential role of the PKC signaling pathway in *C. acnes*-induced cytokine production. These results are consistent with the involvement of H1Rs in pro-inflammatory molecule production by keratinocytes and in human alveolar epithelial cells by *Legionella pneumophila* [31,44]. No interaction between *C. acnes* and PKC has ever been demonstrated. However, *C. acnes* influences lipogenesis by controlling the activity of the diacylglycerol acyltransferase, thereby increasing the amount of diacylglycerol (DAG), which is known to activate PKC [45,46]. More investigations will be necessary to evaluate the impact of meclozine on the PKC pathway.

There is currently no animal model reproducing inflammatory acne. However, the intradermal injection of live *C. acnes* bacteria can lead to the development of an inflammatory skin response [47,48]. In this study, we used an inflammatory score encompassing the thickness, redness and peeling on the ear, and, thus, reflecting all the consequences of *C. acnes*-induced inflammation. Indeed, the ear thickness tended to decrease rapidly while redness and pustules remained. This discrepancy with results from other studies using this model could be due to the mouse strain used here (BALB/c), different from other studies (ICR) [48]. The ear inflammation observed after *C. acnes* injection was characterized by an increase in score one to two days after injection. Redness and peeling mostly appeared more than three or four days after the injection. We first prepared an in-house gel by combining meclozine with a non-ionic solubilizer incorporated into Vaseline. We found that meclozine was effective and acted in a concentration-dependent manner, with a minimal active concentration of 1% significantly decreasing *C. acnes*-induced inflammation, by 14.3%. Interestingly, the pharmacological formulation of meclozine at a concentration of 1% decreased ear inflammation by 26.7%. The difference in efficacy of the two preparations may be due to better solubilization and homogenization of meclozine in the pharmacological preparation.

We assessed the action of meclozine in an ex vivo model based on the *C. acnes* stimulation of human skin explants, as skin is composed of a heterogeneous cell population including several immune cells involved in the innate immune response. We found that *C. acnes* induced the transcriptional and translational production of IL-1β, TNF-α and GM-CSF, which decreased after meclozine pre-treatment. We were not able to evaluate the production of CXCL8/IL-8 in this model, due to the large amounts of SVF in the culture medium. GM-CSF is produced by hematopoietic and non-hematopoietic cells, such as fibroblasts, endothelial and epithelial cells, in response to stimulation. Along with CXCL8/IL-8, GM-CSF is implicated in the recruitment of macrophages to the site of inflammation [49], and the presence of a “CSF network” involving the proinflammatory cytokines IL-1, TNF and GM-CSF has been proposed as part of a positive feedback loop between macrophages and their neighboring cell population, contributing to the maintenance of chronic inflammation [50,51]. Moreover, in vitro experiments have shown that *C. acnes* induced the production of TNF-α, IL-1β, IL-1α and GM-CSF by keratinocytes and monocytes [20,25,52]; the skin explant may be an interesting tool to evaluate the effect of molecular compounds on *C. acnes*-induced inflammation.

Having shown that meclozine could decrease *C. acne*-induced inflammation in vitro, in vivo and ex vivo, we investigated its efficacy in humans with mild-to-moderate acne. To date, only the metabolism and kinetics of meclozine in oral formulations for humans have been studied, and CYP2D6 appears to be the principal hepatic enzyme metabolizing this molecule [53]. However, toxicological tests performed in accordance with the requirements of cosmetological standards showed that meclozine was not toxic. Using a pharmacological 2% meclozine gel in a pilot study, we were able to decrease ASI score by 20.1% (*p* < 0.001) in the treated group, whereas this score decreased only by 8.9% (non-significant) in the control group.

Standard topical treatments for mild-to-moderate acne include topical retinoids, benzoyl peroxide (BPO), azelaic acid, antibiotics and combinations of these agents. The most common adverse events reported are local skin irritation, which seems to be most frequent with BPO and retinoids [26,54]. However, topical antibiotic use had led to an increase in antibiotic resistance in *C. acnes* strains and to the modification of the cutaneous microbiota, contributing to the introduction of a selective pressure leading to antibiotic resistance and the selection of *C. acnes* phylotype IA_1_ strains associated with acne [55,56]. For these reasons, the use of antibiotics is of limited use as long-term and monotherapy regimens in acne [57]. Therefore, the use of BPO alone or in combination with topical retinoid is more recommended as an alternative treatment [58]. It was reported that BPO at 2.5 to 5% applied for 8 to 12 weeks was able to reduce the mean percent difference of total lesion count by 15 to 30% between the treated and the control groups, but was associated with the presence of adverse events during all the studies [59]. Indeed, BPO is associated with several adverse effects, such as irritation, erythema, scaling and bleaching, reducing its therapeutic value by low adherence of patients [60]. In our study, the difference of ASI between the treated and the control groups was 11.2% in ITT and 12% in PP. Importantly, no adverse events at all were reported during the 12-week treatment. In term of efficacy, meclozine appears to have an activity comparable to that of BPO but the good tolerance of meclozine observed here contrasts with the frequent adverse effects reported with BPO. Moreover, it has been reported that H-1 antagonist compounds have the ability to reduce sebum production in sebocytes as well as showing anti-bacterial activity [61,62], which could contribute to reducing acne lesions. This preliminary clinical trial provided proof-of-concept for the use of meclozine in topical application for acne. Its major limitation was the small number of participants enrolled but it could be used to support further larger clinical trials comparing the efficacy and tolerance of meclozine with BPO. Furthermore, as the combination of BPO with retinoid, such as adapalene, improved treatment efficacy [63,64], it will be valuable to test meclozine in combination with adapalene to evaluate its usefulness in replacing BPO in preparations as first-line treatment for mild-to-moderate acne.

## 5. Conclusions

In summary, we identified, by a drug repurposing strategy, meclozine dihydrochloride as a new anti-inflammatory compound. We demonstrated this new property through several *C. acnes*-induced inflammations models in vitro, in vivo and in ex vivo human skin explant. Preliminary, proof-of-concept clinical study realized under cosmetical standards on 60 volunteers for 12 weeks showed that 2% meclizine gel reduced significantly the mean percent difference of acne lesion by 12% in the treated group compared to the control group. No adverse events were reported. This new therapeutic class molecule appears to have comparable efficacy to BPO, used in midle-to-moderate acne treatment and associated with several adverse effects. This could be a useful pilot study that could be used to support the evaluation of the efficacy and safety of meclozine, compared to BPO, and its combination with retinoid in further larger clinical trials. Moreover, as the imbalance of skin microbiota is linked to the presence of acne lesions, the design of clinical trial will be of utmost importance to evaluate the impact of meclozine on the presence of the different microorganisms at the surface of the skin, before and after treatment.

## Figures and Tables

**Figure 1 biomedicines-10-00931-f001:**
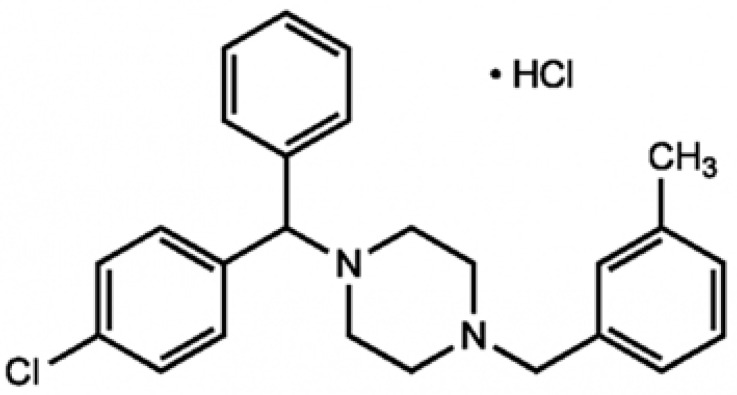
Chemical structure of meclozine. Meclozine hydrochloride or 1-(p-chloro-alpha-phenylbenzyl)-4-(m-methyl-benzyl)—piperazine dihydrochloride monohydrate.

**Figure 2 biomedicines-10-00931-f002:**
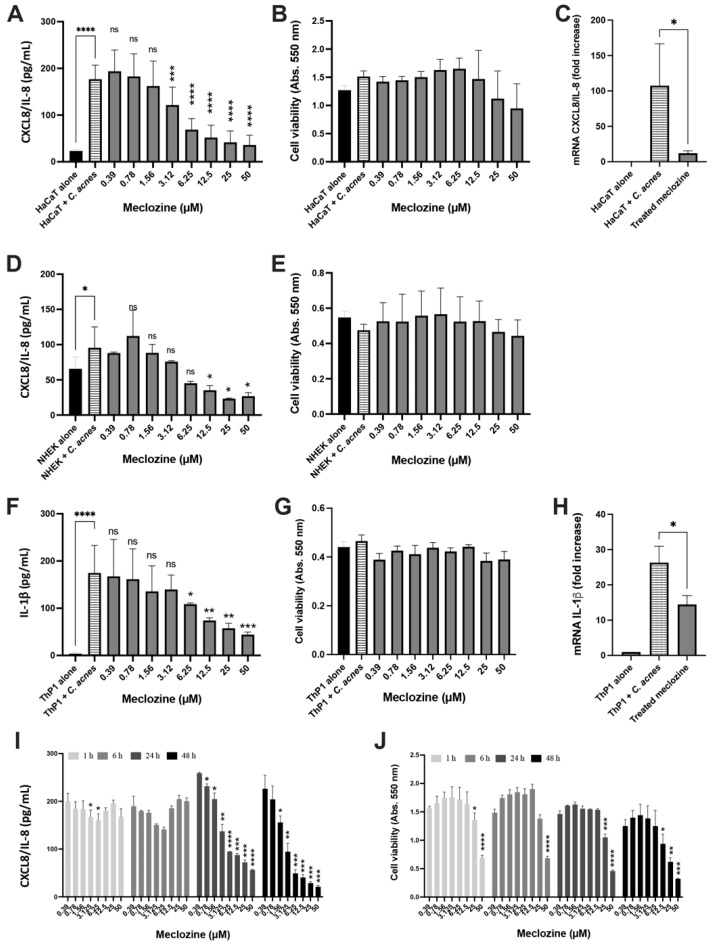
Meclozine inhibits the CXCL8/IL-8 and IL-1β productions at the translational and transcriptional levels. (**A**–**C**) HaCaT cell, (**D**,**E**) primary keratinocyte (NHEK), (**F**–**H**) ThP-1 cell were pre-treated for 24 h with meclozine at concentrations ranging from 0.39 to 50 µM and then stimulated with *C. acnes* 6919 (gray bar). Control experiments were conducted with untreated and unstimulated cells (black bar) and with cells stimulated with *C. acnes* 6919 only (horizontal line bar); *n* = 4. (**C**,**H**) Total RNA was extracted from HaCaT and ThP-1 cells pre-treated with 10 µM meclozine and mRNA levels of CXCL8/IL-8 and IL-1β were quantitated by real-time RT-PCR and compared with GAPDH mRNA level (used as control) and are expressed as fold-change; *n* = 3. (**I**,**J**) HaCaT cells were pre-treated with meclozine at concentrations ranging from 0.39 to 50 µM for 1 h (dark bar), 6 h (dark gray bar), 24 h (middle gray bar), and 48 h (light gray bar), and then stimulated by *C. acnes* 6919 for 18 h. Measurement of CXCL8/IL-8 and IL-1β productions were realized by ELISA and cytotoxicity was determined by the MTT assay, *n* = 3. Data are means ± S.D. of separate experiments. Statistical significance was indicated by * (*p* < 0.05), ** (*p* < 0.01), *** (*p* < 0.001), and **** (*p* < 0.0001), respectively. ns: non-significant.

**Figure 3 biomedicines-10-00931-f003:**
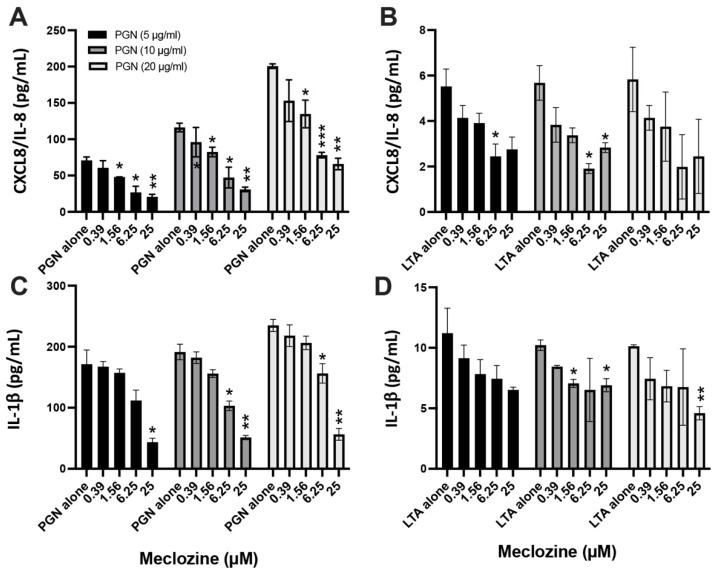
Meclozine inhibits CXCL8 and IL-1β productions in PGN and LTA stimulated keratinocytes and monocytes. HaCaT cells (**A**,**B**) and ThP1 cells (**C**,**D**) were pre-treated for 24 h with meclozine at concentrations ranging from 0.39 to 25 µM and then stimulated with PGN and LTA at 5 (dark gray bar), 10 (middle gray bar), and 20 µg/mL (light gray bar). Control experiments were conducted with cells stimulated with PGN and LTA only. Measurement of CXCL8/IL-8 and IL-1β productions were realized by ELISA. Data are means ± SEM of three separate experiments. Statistical analysis was done by using PGN and LTA alone for each concentration used as a reference and are indicated by * (*p* < 0.05), ** (*p* < 0.01), and *** (*p* < 0.001) respectively.

**Figure 4 biomedicines-10-00931-f004:**
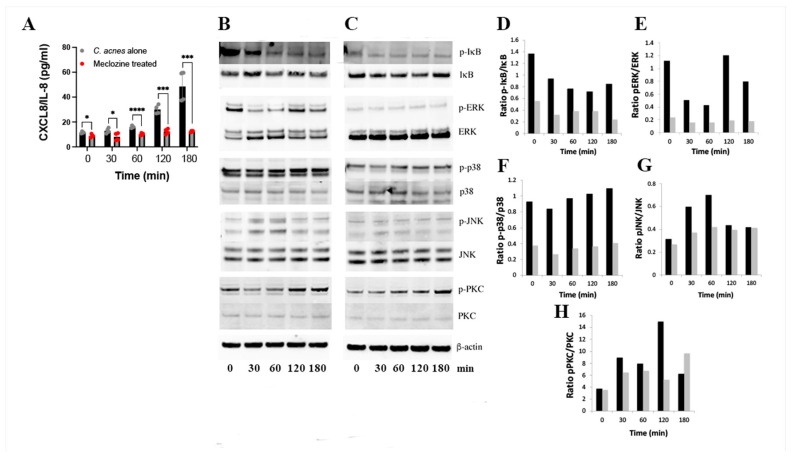
Meclozine inhibits inflammatory signaling pathways. HaCaT keratinocytes were pre-treated for 24 h with 10 µM meclozine and then stimulated with *C. acnes* (MOI of 15) for 30, 60, 120 and 180 min. (**A**) CXCL-8/IL-8 production was measured by ELISA in culture supernatant. Data represent mean ± SEM, *n* = 4. Representative blot analysis of relative phosphorylation of proteins (**B**) after C. acnes stimulation and (**C**) after meclozine pre-treatment and *C. acnes* stimulation. (**D**–**H**) Quantitative analysis of immunoblotting with *C. acnes* stimulation alone (dark bar) and meclozine pre-treatment followed by *C. acnes* stimulation (gray bar). Statistical significance is indicated by * (*p* < 0.05), *** (*p* < 0.001), and **** (*p* < 0.0001), respectively.

**Figure 5 biomedicines-10-00931-f005:**
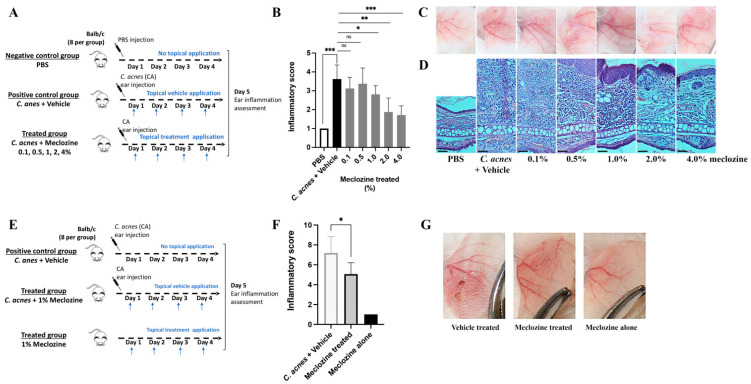
In vivo meclozine activity in *C. acnes*-stimulated ear mouse model. (**A**,**E**) Experimental setup of the in vivo training model used. BALB/c mice ears were injected with PBS (negative control) or stimulated with *C. acnes* suspension (1.5×10^6^ CFU/20 µL). Both ears were topically treated with 30 mg of vehicle (positive control) or with (**A**–**D**) 30 mg of in-house meclozine gel formulation at 0.1 to 4% meclozine; and (**E**–**G**) 30 mg of pharmaceutical gel formulation at 1% meclozine. (**B**,**F**) Inflammatory score, including ear thickness, redness and peeling, was measured for each ear. Each bar represents the mean ± SEM, *n* = 8 biologically independent mice. (**C**,**G**) Pictures of ears. (**D**) Histological analysis of ears. The Mann–Whitney test was used to detect significant differences between groups with ns: non-significant; * *p* ≤ 0.05; ** *p* ≤ 0.01; *** *p* ≤ 0.001. Scale bar = 50 μm.

**Figure 6 biomedicines-10-00931-f006:**
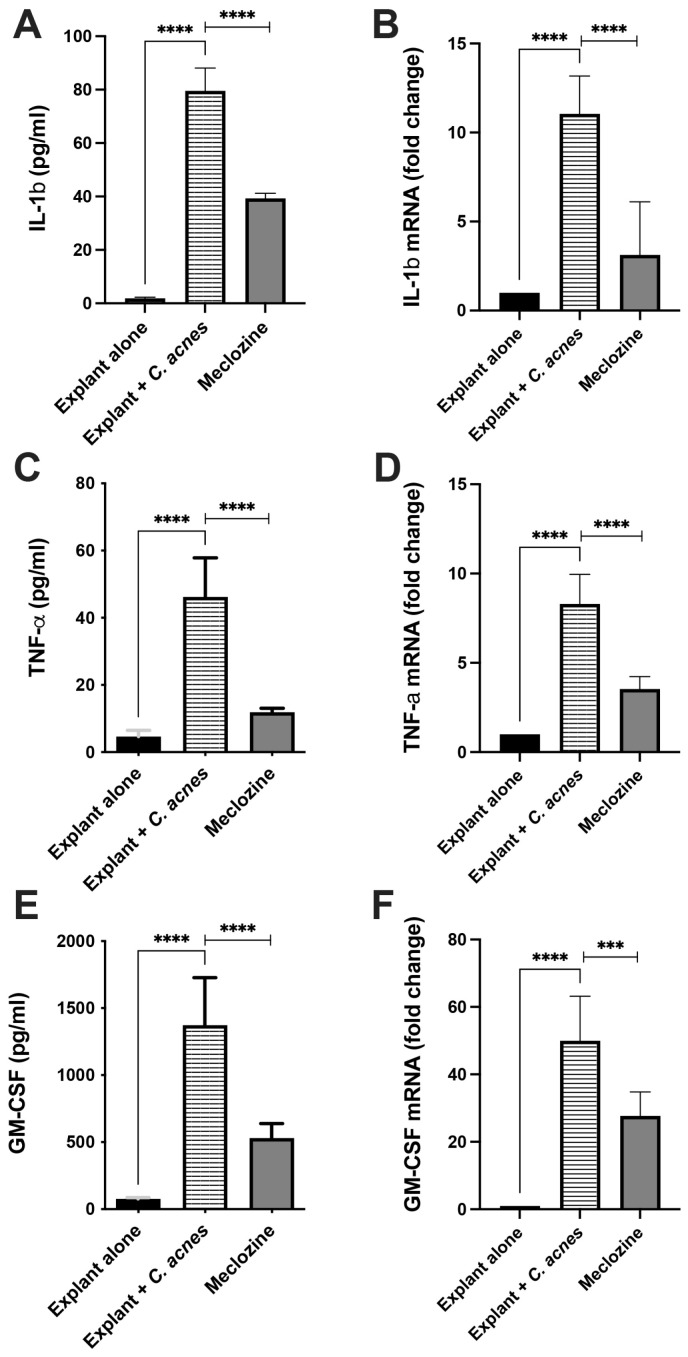
Ex vivo meclozine activity. Human skin explants were left untreated and unstimulated (explant alone), stimulated by *C. acnes* 6919 (O.D._600nm_ = 1.0) (explant + *C. acnes*), and pre-treated with 15 µM meclozine for 48 h and then stimulated by *C. acnes* for 24 h. (**A**,**C**,**E**) IL-1β, TNF-α and GM-CSF productions were measured by ELISA in culture supernatant, respectively. (**B**,**D**,**F**) Total RNA was extracted and IL-1β, TNF-α and GM-CSF mRNA levels were determined by real-time RT-PCR and compared with GAPDH mRNA level (used as control), and are expressed as fold-change, respectively. Data are means ± SEM, *n* = 3. Statistical significance is indicated by *** (*p* < 0.001), and **** (*p* < 0.0001), respectively.

**Figure 7 biomedicines-10-00931-f007:**
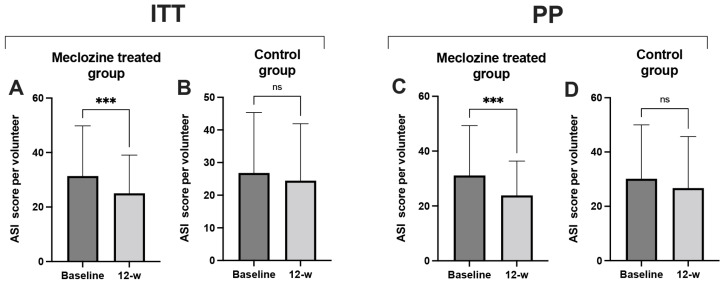
Clinical evaluation of meclozine in acne lesions in human. Double-blind, randomized, placebo-controlled clinical evaluation of 2% meclozine gel was evaluated over 12 weeks in 60 volunteers as described in Materials and Methods. Acne severity index (ASI) was determined at baseline and after 12 weeks treatment in the intention to treat (ITT) and per-protocol (PP) approaches. (**A**,**C**) Treated group and (**B**,**D**) control group. The Wilcoxon signed-rank test was used to detect significant differences between groups with ns: non-significant; *** *p* ≤ 0.001.

**Table 1 biomedicines-10-00931-t001:** Demographic characteristic of volunteers.

	Treated Group	Control Group
Age, n	30	30
Mean (S.D.)	27 (5.34)	27 (6.43)
Median	25	27
(Min, max)	(19, 38)	(19, 39)
Age categories		
>18	1	5
20–30	20	14
30–40	9	11
Sex, n (%)		
Man	14 (46.7)	6 (20)
Woman	16 (53.3)	24 (80)

## Data Availability

Data contained within the article and the original data that support the findings of the present study are available from the corresponding author upon reasonable request.

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
