# Peer review of "A New Topical Candidate in Acne Treatment: Characterization of the Meclozine Hydrochloride as an Anti-Inflammatory Compound from In Vitro to a Preliminary Clinical Study"

_biomedicines, 2022, doi:10.3390/biomedicines10050931_

Round 1

Reviewer 1 Report

The work is innovative because the use of an antihistamine drug in acne is new. The job is well done. You used Visia's in vivo clinical exam on volunteers during the 12 weeks of the study.
So I ask to add some data
that you can easily determinate by Visia: 1) The acquisition of facial images with VISIA allows the detection of the quantity of porphyrins and their distribution.
It would be appropriate to add this data, because acne is often associated with an imbalance of the skin microbiota.
2) Also if you have a sebumeter, it's interresting sebum measurement before and after the treatment.

Author Response

Reviewer #1

The work is innovative because the use of an antihistamine drug in acne is new. The job is well done.

You used Visia's in vivo clinical exam on volunteers during the 12 weeks of the study.
So I ask to add some data that you can easily determinate by Visia:

1) The acquisition of facial images with VISIA allows the detection of the quantity of porphyrins and their distribution. 
It would be appropriate to add this data, because acne is often associated with an imbalance of the skin microbiota. 
2) Also if you have a sebumeter, it's interresting sebum measurement before and after the treatment.  

Answer:

We acknowledge that microbiota imbalance and sebum production were at of interest in in vivo acne clinical exam, however during our study we didn't performed these measurements.

Reviewer 2 Report

I have read the paper with interest.

Although not examined in this study, H1 antihistamines have been reported to enhance the efficacy of antibacterials and reduce sebum production which may also be its mode of action in acne. This should be briefly mentioned in discussion.

Author Response

Reviewer #2

I have read the paper with interest.

Although not examined in this study, H1 antihistamines have been reported to enhance the efficacy of antibacterials and reduce sebum production which may also be its mode of action in acne. This should be briefly mentioned in discussion.

Answer:

  • The manuscript has been carefully read
  • We agreed that the antihistaminic compounds have these properties, therefore we added this sentence in the Discussion section (line 641) and added references accordingly:

Moreover, it has been reported that H-1 antagonist compounds have the ability to reduce sebum production by sebocytes as well as to have anti-bacterial activity (Pelle et al., 2008; Boyd et al., 2021) which could contribute to reduce acne lesions.